# Automatic real-time prostate detection in transabdominal ultrasound images

**Tiziano Natali**[1]      T.NATALI@NKI.NL
**Mark Wijkhuizen**[1]      M.WIJKHUIZEN@NKI.NL
**Liza Kurucz**[1]      L.KURUCZ@NKI.NL
**Matteo Fusaglia**[1]      M.FUSAGLIA@NKI.NL
**Pim J. van Leeuwen**[1]      PJ.V.LEEUWEN@NKI.NL
**Theo J.M. Ruers**[1,2]      T.RUERS@NKI.NL
**Behdad Dashtbozorg**[1]      B.DASHT.BOZORG@NKI.NL

[1] *Image-Guided Surgery, Netherlands Cancer Institute, Amsterdam, The Netherlands*
[2] *Faculty of Science and Technology, University of Twente, Enschede, The Netherlands*

## Abstract

Prostate cancer is the second most common form of cancer in men, thus easily accessible early diagnostic tools are of vital importance. Transabdominal ultrasound is an inexpensive, non-invasive, and accessible imaging modality. However, generated images are challenging to interpret and require expert clinicians to be assessed. Therefore, we propose a DL model for real-time automatic detection of the prostate for guidance to inexpert operators. Results show that the proposed model has similar performance to state-of-the-art models, achieving mean average precision at 0.50 of 0.95 and queries per second of 993, indicating possible clinical application and possible improvement with more extensive pretraining strategies.

**Keywords:** CAD, Detection, Prostate, Transformer, CNN, Real-Time, Ultrasound imaging.

## 1. Introduction

Transrectal ultrasound (TRUS) is the standard imaging modality when assessing prostate volume as an early sign of prostate cancer. TRUS is a relatively inexpensive modality and provides continuous imaging without inducing ionizing radiations, with excellent resolution and soft tissue contrast. On the other hand, transrectal procedures are characterized by patient discomfort due to their invasiveness, necessitating to be operated by expert clinicians. At the same time, prostate cancer is the second most common cancer among men(Bray et al., 2024), hence accessible early diagnostic methods are necessary for early detection and management of the disease.

Implementation of trans-abdominal ultrasound (TAUS) prostate scans to aid the early diagnosis of prostate cancer has been the aim of recent studies (de Vos et al., 2023). This procedure solves the problems related to patient discomfort and speeds up the total acquisition time. However, the assessment of prostate volume from TAUS acquisitions remains challenging. Acquired prostate TAUS images are often affected by low signal-to-noise ratio and shadow artifacts caused by the nearby pelvic bone, hindering the locating of the prostate and assessment of its borders for volume estimation. Consequently, expert clinicians are still required during acquisitions.

Deep learning models have shown to be able to achieve similar results to expert human operators in many medical imaging fields (Anwar et al., 2018), and could achieve similar results in prostate detection.

To address this issue, in this study we propose an automatic algorithm for real-time prostate detection, aimed at assisting inexpert, untrained radiologists or less-skilled operators when acquiring TAUS

images of the prostate. The queries per second demonstrating promising results in real-time automatic detection of the prostate in transabdominal US images.

## 2. Methods

The dataset used in this study consists of 7500 ultrasound frames from 55 video acquisitions of 55 patients, with ground truth bounding boxes around prostate locations provided by an expert clinician. This has been first divided at the patient level into train and test sets, following a 90% - 10% ratio respectively, then 10% of the train set was used as the validation set. Images were grayscale with a size of $750 \times 750$ pixels, which were downsized to $640 \times 640$ to match the models' input sizes.

The proposed network has been shaped as a detection transformer (based on the model presented by Carion *et al.* (Carion et al., 2020)) since US acquisitions are sequences of images, and the nature of transformers can better take advantage of it. The network presents a shallow architecture, summing to a total of about 3 million parameters. Additionally, for the sake of comparing with a state-of-the-art technique, the YOLOv8 model in its nano format was trained on the same data.

Both models have been trained in two modalities: i) randomly initialized weights and no data augmentation (NanoDeTr-scratch and YOLOv8-scratch); ii) pre-trained weights on the ImageNET for NanoDeTr and the COCO dataset for YOLOv8 plus data augmentation in the form of horizontal flipping, intensity shift, rotation, translation, compression, and Gaussian noise. Both models were trained with an Adam optimizer and binary cross-entropy plus generalized intersection over union loss (Rezatofighi et al., 2019).

The metrics used for the evaluation of the models were mean average precision (mAP), and mean average precision when the prediction intersection over union (IoU) threshold is set at 0.50 (mAP@50) as well as at 0.75 (mAP@75). The running speed of the models has been evaluated in the form of queries per second (QPS) using the TensorRT library. The machine used for the training and testing of the models was equipped with an Intel Xeon W-1250P CPU, Nvidia Geforce RTX 4060ti 16GB, and 32GB of RAM memory.

## 3. Results

The results summarized in Table 1 shows that the YOLOv8 model is the one with highest values for all metrics. When it comes to mAP@50,the results between NanoDeTr and YOLOv8 are similar, only differing by 0.03. The same trend goes for NanoDeTr-Scratch and YOLOv8-Scratch, where the models differ by only 0.01. On the hand, when comparing the NanoDeTr and YOLOv8 models for mAP@75, these present a difference of 0.12, that increase to 0.21 when comparing NanoDeTr-Scratch and YOLOv8-Scratch. A comparison between the results for mAP@75 of NanoDeTr-Scratch and NanoDeTr takes great advantage of the pretraining on ImageNet, showing an improvement of 0.12, while YOLOv8 only improves by 0.03 from YOLOv8-Scratch. The results of the two models in terms of QPS show NanoDeTr outperforming YOLOv8 by about 80.

Figure 1 demonstrates the models' results on some exemplary images from the test set. From a qualitative assessment of the output on the test set, it appeared that NanoDeTr output on average larger boxes but still around the correct area, and that the vast majority of the missed instances are at the prostate margins (i.e., when the US slices the prostate near its border).

Table 1: Trained models and their results on the test set for mAP, mAP@50, mAP@75 and QPS.

| Model | mAP | mAP@50 | mAP@75 | QPS |
|---|---|---|---|---|
| YOLOv8-Scratch | 0.65 | 0.95 | 0.81 | $915 \pm 20$ |
| NanoDeTr-Scratch | 0.55 | 0.94 | 0.60 | $993 \pm 15$ |
| YOLOv8 | 0.68 | 0.98 | 0.84 | $915 \pm 20$ |
| NanoDeTr | 0.60 | 0.95 | 0.72 | $993 \pm 15$ |

Figure 1: Example results for the proposed networks and ground truth as defined in Sec. 2. The left image showcase a significant difference in results of two models, the two images in the middle show small differences, and the one to the right a quite similar results. The YOLOv8 model manages to cover the ground truth with higher IoU across all cases.

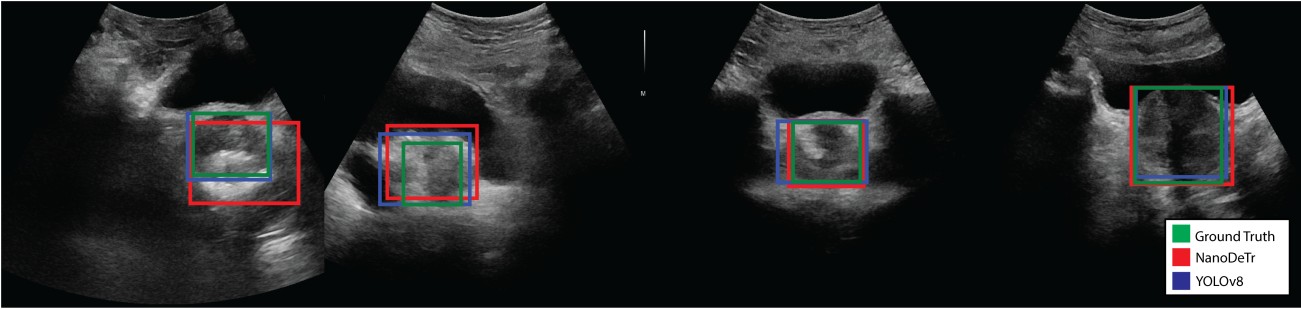

## 4. Discussion and Conclusion

In this study, we presented the NanoDeTr model, a novel detection transformer trained for real-time detection of the prostate from TAUS images, and compared it to YOLOv8 trained for the same task. Both models showed similar performance in mean average performance and queries per second. However, it is worth mentioning that NanoDeTr has a simpler architecture with a significantly smaller number of parameters compared to YOLOv8.

The YOLOv8 model overall achieved slightly better results than NanoDeTr, where the slight underperformance of NanoDeTr might be explained by the following factors. It is renowned that vision transformers can better generalize on larger datasets than CNNs (Maurício et al., 2023), hence the lower scores for NanoDeTr-Scratch with respect to YOLOv8-Scratch. This is also supported by results in Table 1, which show a slight improvement between YOLOv8-Scratch and YOLOv8 across all metrics, while there is a substantial improvement for NanoDeTr. This hints at the fact that transformers can better take advantage of pretraining than CNNs. The underperformance of NanoDeTr vs. YOLOv8 might be related to the fact that the first is pre-trained on ImageNet, while the latter is trained on the COCO dataset. This implies that only the decoder block of NanoDeTr is pre-trained, while in the case of YOLOv8 also the detection and classification blocks are pre-trained. Therefore, we believe that by improving the training of NanoDeTr it will be possible to achieve similar results to YOLOv8 with even higher QPS.

With the achieved results, we can conclude that the presented models are suited for real-time guidance and aid during prostate TAUS acquisitions for untrained radiologists and inexpert users.

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
