# OpenReview forum: "Automatic real-time prostate detection in transabdominal ultrasound images"
_MIDL.io/2024/Short_Papers — MIDL 2024 Short Papers_

### Official Review · Reviewer_rtMy · 2024-04-17

**Confidence:** 5
**Final Rating:** 4

**Review:**

Guiding novice users in interpreting ultrasound data is crucial as this imaging modality becomes increasingly accessible outside hospital environments. The authors propose a prostate detection framework, though its novelty is somewhat limited due to previous similar methods being explored. Results vary significantly between challenging scans and high-quality scans, but the work remains intriguing and merits presentation at the MIDL conference.

---

### Decision · Program_Chairs · 2024-04-26

Accept